# Investigation of Tactile Illusion Based on Gestalt Theory

**Hiraku Komura** [1,*], **Toshiki Nakamura** [2] **and Masahiro Ohka** [2]

1    Faculty of Engineering, Kyushu Institute of Technology, 1-1 Sensui-cho, Tobata-ku,
Kitakyushu-shi 804-8550, Japan

2    Graduate School of Informatics, Nagoya University, Furo-cho, Chikusa-ku, Nagoya 464-8601, Japan;
toshik.nakamura@teijin.co.jp (T.N.); ohka@i.nagoya-u.ac.jp (M.O.)

*    Correspondence: komura@cntl.kyutech.ac.jp

**Abstract:** Time-evolving tactile sensations are important in communication between animals as well as humans. In recent years, this research area has been defined as "tactileology," and various studies have been conducted. This study utilized the tactile Gestalt theory to investigate these sensations. Since humans recognize shapes with their visual sense and melodies with their auditory sense based on the Prägnanz principle in the Gestalt theory, this study assumed that a time-evolving texture sensation is induced by a tactile Gestalt. Therefore, the operation of such a tactile Gestalt was investigated. Two psychophysical experiments were conducted to clarify the operation of a tactile Gestalt using a tactile illusion phenomenon called the velvet hand illusion (VHI). It was confirmed that the VHI is induced in a tactile Gestalt when the laws of closure and common fate are satisfied. Furthermore, it was clarified that the tactile Gestalt could be formulated using the proposed factors, which included the laws of elasticity and translation, and it had the same properties as a visual Gestalt. For example, the strongest Gestalt factor had the highest priority among multiple competing factors.

**Keywords:** tactileology; tactile gestalt; principle of prägnanz; law of closure; formulation; psychophysics; velvet hand illusion; dot-matrix display; texture sensation

## 1. Introduction

The human tactile sense plays an important role not only in sensing the external environment but also in communication. Tactile researchers have focused on the former, including efforts to understand how humans detect an object in their environment and perceive it as a subjective tactile sensation. In this research, various types of actuators have been developed for tactile displays [1]. In addition, different psychophysical experimental methods for evaluating human subjective sensations have been developed [2], along with some mathematical formulas to describe the relationship between psychological quantities and physical quantities, such as Stevens' power law and the Weber–Fechner law. On the other hand, there has been little systematic research on communication via the tactile sense. When considering communication via the sense of touch, it seems necessary to consider the time evolution of the tactile sensation. Thus, the term "tactileology" was suggested by Suzuki to define the new interdisciplinary research field of time-evolutional tactile sensation [3]. Tactileology focuses on the relationship between the time-evolving skin deformation and not only the perception of object property but also the sensibility such as "comfortableness". To advance this research field, this study attempted to clarify the effects of a time-varying input on a human.

A tactile score [4] and humanitude [5] are some of the techniques currently in use that utilize a time-varying tactile sensation. The tactile score was developed for the massage field. To obtain tactile scores, the massage techniques of therapists are converted into force, time and touch position data by using improved music scores, which can be recorded and reproduced at any time. Humanitude was developed to help dementia patients

relax through communication by using a combination of the visual, tactile and auditory senses. It is very interesting to note that various stimuli that change over time, especially tactile stimuli, relieve stress or change a person's mood. We believe that elucidating this mechanism will be important for humans living in a stressful society.

To investigate the time-varying tactile sensation, this study focused on the Gestalt theory [6] because the feature of an auditory Gestalt could be used as a reference. The gestalt theory was suggested by Ehrenfels et al. based on the auditory and visual senses. According to "A source book of gestalt psychology" [7], the fundamental formula of the Gestalt theory might be expressed as follows: there are wholes, the behavior of which is not determined by that of their individual elements, but where the part-processes are themselves determined by the intrinsic nature of the whole. In other words, the wholes cannot be expressed by a simple sum of elements and elucidating the mechanism by which the elements are integrated is the research objective. Ehrenfels noted that in music, the melody was preserved even if the pitch was raised or lowered as a whole. Moreover, the recognition of a shape is preserved even if the size of the shape changes. Studies on auditory and visual Gestalts have been progressing, and many rules have been systematically summarized as the principle of Prägnanz. In this way, the Gestalt theory can deal with the mechanism for integrating many elements, which current science and technology are not good at handling. For this reason, it was assumed that this theory could also handle the integration of tactile sensations that change over time.

To consider tactileology based on a tactile Gestalt, this study explored tactile illusion phenomena such as the velvet hand illusion (VHI) [8] because it was assumed that the VHI is caused by a tactile Gestalt mechanism. In the VHI, a smooth sensation rather than the wire sensation is induced when the net part of a tennis racket is rubbed back and forth between the palms of the hands. The VHI is not generated under a one-wire condition but is generated under the condition of more than one wire [9,10]; thus, this illusion is caused by the integration of multiple stimuli.

In this research, a philosophical goal of this study was to clarify tactileology using the tactile Gestalt theory. This is because the clarification of tactileology was considered to contribute towards the understanding of fundamental nature of human's tactile sense. To achieve this goal, the relationship between the tactile Gestalt and VHI was investigated because this illusion is caused by the integration of several stimuli and this illusional sensation is different from that based on the element's material. Two types of experiments were performed to clarify the tactile Gestalt theory. First, a psychophysical experiment was conducted to clarify the relationship between the law of closure, which is one of the principles of Prägnanz, and the appearance of the VHI. In the second experiment, an attempt was made to formulate the VHI intensity and two-line physical stimulation. For instance, when two parallel wires are moved back and forth across the palms of a human's hands with a phase difference, the closed area with wires is stretched, shrunk or translated. This area created by multiple parts was considered the Gestalt. Two new laws were proposed: the law of translation and the law of elasticity. These laws could explain the variable state of the Gestalt and formulate the relationship between the VHI and movement of two parallel wires. Through this research, we aimed to construct a tactile basic theory for an engineering application.

## 2. Gestalt Theory

### 2.1. Gestalt

Gestalt grouping helps humans recognize the information of separate parts as a unit [6]. Moreover, illusion phenomena related to Gestalt grouping have also been discovered. When perceiving an object through visual, auditory and tactile senses, humans recognize a set of information as a unit called a Gestalt. These Gestalts enable us to recognize facial expressions and enjoy listening to music. Depending on the law of the Gestalt, interesting illusion phenomena with respect to the visual and auditory senses have been reported, such as the visual Müller–Lyer illusion [11] and auditory octave illusion [12,13].

## 2.2. Visual Gestalt

A visual Gestalt is an essential mechanism for perceiving various objects in the environment [6]. Without a visual Gestalt, it is not possible to read characters and recognize facial expressions when seeing moving facial parts. The visual Gestalt consists of many factors such as the laws of proximity, similarity, closure and common fate in the Prägnanz principle [8]. Figure 1 describes some of the factors assumed in the Prägnanz principle. As shown in Figure 1a, the law of closure helps the brain perceive the entire appearance of shapes and figures even when one or more of their parts are hidden or completely absent. Figure 1b shows how the law of similarity allows objects of the same shape, color and property to be grouped. In Figure 1c, the law of common fate helps a person recognize objects moving in the same direction as a group.

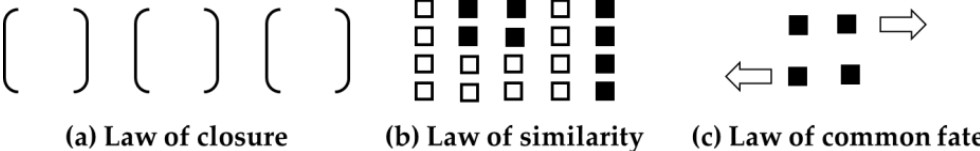

(a) Law of closure   (b) Law of similarity   (c) Law of common fate

**Figure 1.** Principle of Prägnanz in Gestalt theory. (**a**) Law of closure, (**b**) Law of similarity, and (**c**) Law of common fate are factors of principle of Prägnanz.

Figure 2 shows the various visual illusions related to the visual Gestalt. Figure 2a demonstrates the Kanizsa triangle, in which the differences in brightness are perceived as contour lines, even though there is no variation in brightness or color along these perceived contour lines [14]. Figure 2b shows the Müller–Lyer illusion, in which the subjective line length changes depending on the direction of the wings at both ends, even though the length of the straight line is the same [11]. Figure 2c illustrates the Zellner illusion, in which the subjective parallelism is lost as a result of the influence of the surrounding lines, even though these multiple straight lines are lined up in parallel [15]. Specifically, the perception is caused not only by actual physical stimuli but also by the grouping process. Moreover, it is obvious that when humans perceive objects, it is insufficient to consider only the local sensations input to their sensory receptors; it is also necessary to understand the integration process completely.

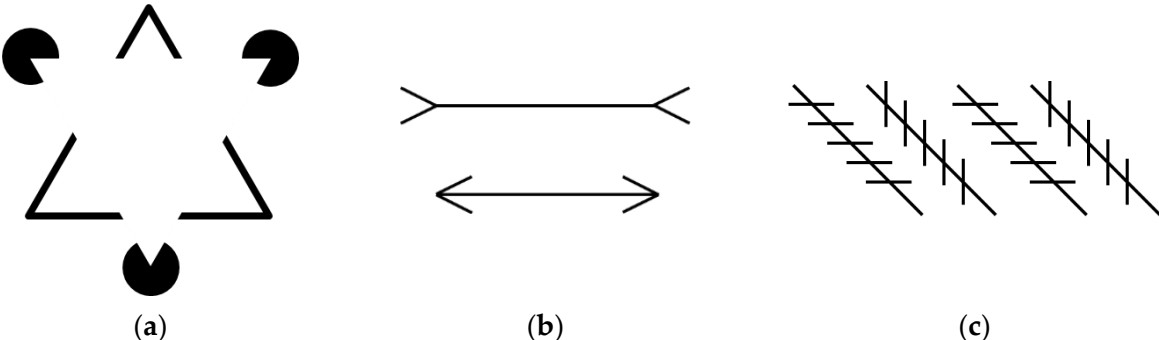

(a)                           (b)                           (c)

**Figure 2.** Visual illusion phenomena based on Gestalt theory. (**a**) Kanizsa triangle, (**b**) Müller–Lyer illusion, (**c**) Zellner illusion.

## 2.3. Auditory Gestalt

Gestalts also exist in the auditory sense [16]. Humans recognize consecutive sounds as melodies. This is because temporal changes in sound and frequencies are grouped and recognized as an auditory Gestalt. For example, several different sounds that start and end simultaneously are recognized as a Gestalt because of the law of common fate. In addition, sounds with similar frequencies and similar lengths are recognized as a Gestalt because of the laws of proximity and similarity, respectively. The basic rule of an auditory Gestalt corresponds to the principle of Prägnanz, which is summarized by the visual Gestalt.

However, in the case of an auditory Gestalt, because the frequency has priority over the space (the position of the sound source), unique illusion phenomena such as the illusion of scale [17,18] or octave illusion [12,13] are invoked.

### 2.4. Tactile Gestalt

Even though tactile Gestalts also exist, humans are not conscious of these tactile Gestalts in daily life because the tactile sense is used to recognize a part of a whole object through tactile motion. It is assumed that the sense of a dent perceived in the Fishbone illusion may be related to the subjective contour in the tactile Gestalt, and this feature is the same as that seen in vision. However, the features of a tactile Gestalt are different from those of a visual Gestalt. For instance, a person might be conscious of the Gestalt when invoking the VHI. It should be noted that a texture sensation is perceived in the tactile Gestalt.

In the tactile recognition mechanism, a texture sensation is generated by integrated signals from the five systems of mechanoreceptors [19] (Meissner's corpuscles, Pacini corpuscles, Merkel cells, Ruffini endings and free nerve endings) in the central nervous system. When the signals from these five receptors are integrated, the feeling of touch is perceived, which comprises graphic information as well as a texture sensation invoked by the tactile Gestalt (Figure 3).

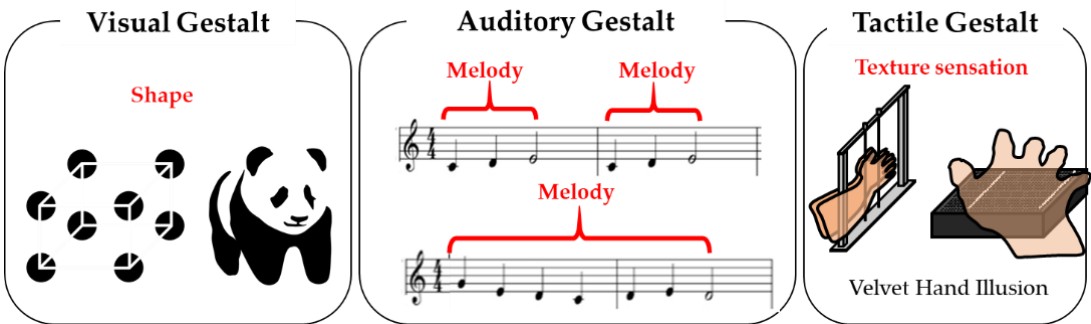

**Figure 3.** Gestalt features in each modality. The shape from parietal parts is recognized in the visual Gestalt. A melody is recognized in the auditory Gestalt when the frequencies close to each other on the time axis are grouped together. A smooth sensation is invoked in the tactile Gestalt.

Moreover, a texture sensation seems to be caused by the intrinsic law of tactile information processing. For example, two factors are related to the VHI based on the principle of Prägnanz. In the VHI stimulus, two parallel wires (lines) are moved back and forth (reciprocated) across the palms of a human's hands. It is assumed that this stimulus can be regarded as a situation in which the law of closure (Figure 1a) and the law of common fate (Figure 1c) hold simultaneously. Thus, the expectation in this study was that the tactile sense mechanism could be more thoroughly understood based on the relationship between the VHI and tactile Gestalt.

## 3. Velvet Hand Illusion

The VHI is a tactile illusion phenomenon in which a smooth sensation is induced, and it is known to occur in most people. Based on a previous study, many experimental findings regarding the stimulus condition have been reported. Mochiyama et al. found that the strongest illusion was caused by two lines (Figure 4), and they tried to formulate the relationship between the stimulus and VHI's smoothness [8]. The intensity of the VHI with a passive touch is nine times stronger than that with an active touch [9], and the intensity of the VHI can be controlled by the ratio of the distance between the two parallel wires and the stroke of the wire's movement [10]. Moreover, the VHI is induced in two parallel lines formed by the stimulus pins in a dot-matrix display [20]. Furthermore, the tactile sensation that occurs in the VHI can be quantified in terms of common materials [21].

Regarding the physiological aspects, investigations of its neural basis using functional Magnetic resonance imaging (fMRI) and Near-infrared spectroscopy (NIRS) have also been performed [22–24]. Thus, in this study, two types of experiments were conducted to clarify the relationship between the VHI and tactile Gestalt and answer the following questions:

➢ Is the VHI caused by the tactile Gestalt mechanism?
➢ Can the VHI be formulated based on the principle of Prägnanz?

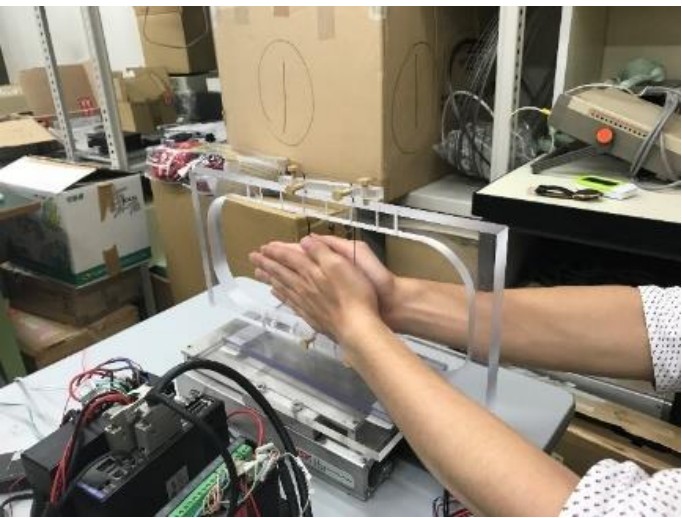

**Figure 4.** Velvet hand illusion display: two stretched parallel wires are moved in reciprocating manner between palms of hands using stepping motor.

To answer the first question, a psychophysical experiment was conducted with a dot-matrix display, and the relationship between the principle of Prägnanz and the VHI occurrence was investigated. To answer the second question, a formulation between the physical and psychological quantities of the VHI was suggested using the principle of Prägnanz and then verified using experimental data.

## 4. Investigation of Relationship between VHI and Tactile Gestalt

To investigate the relationship between the VHI and tactile Gestalt, a tactile display system was developed with a dot-matrix display and used for a psychophysical experiment. This experiment controlled the intensity of the law of closure, which was invoked by the density of pins that made up the line. Each participant's subjective value of VHI intensity were measured as psychological quantities. It was hypothesized that if the law of closure became weak, the VHI would also become weak.

### 4.1. Tactile Display System

The tactile display was developed with a dot-matrix display in which 768 tactile pins were moved up and down by a bimorph piezoelectric ceramic actuator. This system consisted of two refreshable Braille displays (SC10, KGS, Saitama, Japan) [25] with a presentation area of 85 mm × 70 mm, which was appropriate for palm presentation, along with a DIO board (PC–2772, Interface Co., Hiroshima, Japan), PC (OS: Windows 10, 64 bit) and Visual C++, as shown in Figure 5a. The pins were arranged at 2.4 mm intervals, which were narrower than the two-point threshold of the palm of the hand (13 mm) [19,26]. Therefore, the participants could recognize the pins in a row as a line, and this display could provide the VHI stimulus, i.e., two reciprocating parallel lines, as shown in Figure 5b,c.

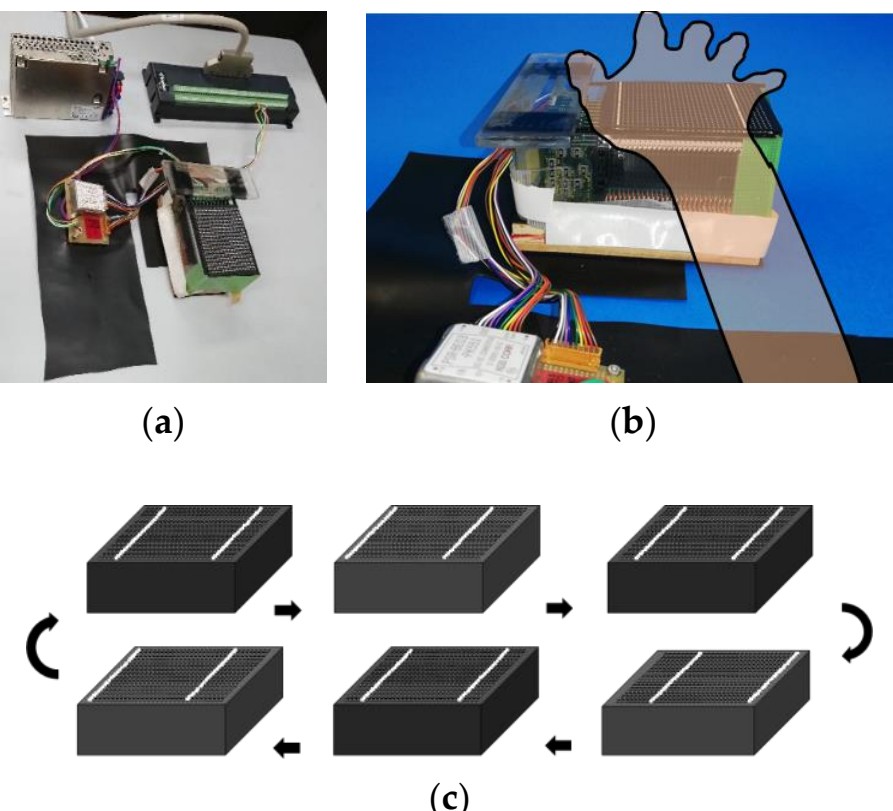

**Figure 5.** Overview of dot-matrix display (upper left: outward appearance; upper right: subject's hand posture during experiment; lower: stimulus in which several pins are continuously moved up and down, creating stimulus such as two parallel lines reciprocating). (**a**) Tactile display, (**b**) Method of use, (**c**) Stimulus in which two parallel lines are reciprocated.

### 4.2. Participants

In this case, 11 male Japanese students participated in the psychophysical experiments. All the participants were fully informed. Their ages ranged between 22 and 25 yr. All the experiments were approved by the ethics committee of the Nagoya University.

### 4.3. Experimental Condition

The purpose of this experiment was to clarify whether there is an important relationship between the VHI and tactile Gestalt. Since the VHI is not induced without movement or when the stimulus condition is a single line, it was assumed that the laws of closure and common fate were the essential factors for VHI induction. Thus, the effects of these factors on the intensity of the VHI were investigated. The basic stimulus conditions were as follows. The line length was 55.2 mm; the distance between the two parallel lines was 60 mm; the stroke was 14.4 mm; and the average speed of the line movement was 80 mm/s. Six types of line conditions were prepared, as shown in Figure 6, in which the interval between the pins was set to 55.2, 26.4, 14.4, 9.6, 4.8 and 2.4 mm. If the law of closure was an essential factor of VHI induction, the VHI would disappear when the participants could no longer recognize the line formed with pins. This experimental trial consisted of six stimulus conditions, and each participant performed this trial four times. In each trial, six types of stimuli were provided to the participants randomly, and they were asked to respond with a "yes" or "no" to questions confirming the VHI appearance.

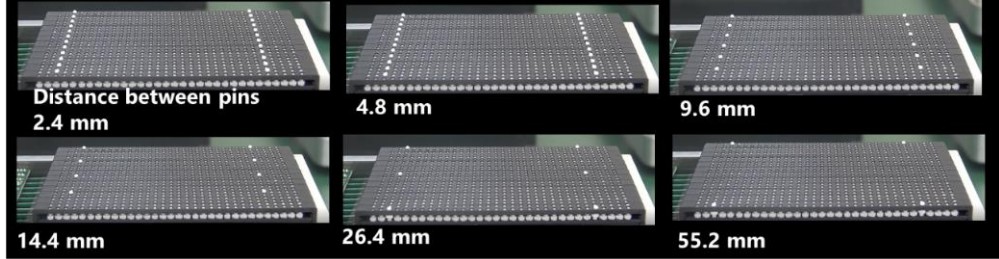

**Figure 6.** Six types of stimulus conditions. The distance between the pins that made up the lines were 2.4 and 55.2 mm under the highest and lowest density conditions, respectively.

### 4.4. Experimental Results and Discussion

Considering the principle of Prägnanz, the VHI might be related to the laws of closure and common fate. Since the VHI is never invoked without line movement, the law of common fate is indispensable. Moreover, it was unclear whether the law of closure was necessary to generate the VHI. Thus, the necessity of the closure law was investigated. The relationship between the law of closure and VHI intensity was clarified by controlling the density of pins along two parallel lines. The probability of invoking the VHI is shown in Figure 7a. An analysis of variance (ANOVA) revealed significant differences between the conditions ($F(5, 50) = 71.2$, $p < 0.001$, $\eta^2 = 0.79$). Subsequently, multiple comparisons were conducted using the Bonferroni correction (IBM SPSS 22.0), and the results revealed significant differences between pin intervals of 4.8 and 9.6 mm and between 9.6 and 14.4 mm. To calculate the threshold of the pin interval to induce the VHI, the probability was converted to a z-score, as shown in Figure 7b. Since a probability of 50% was considered the threshold, which was equivalent to a z-score of zero, a pin interval of 9.34 mm was confirmed to be the threshold. The threshold of the pin interval was less than the two-point threshold for the palm of the hand, which is approximately 13 mm [19]. Therefore, it was clarified that the VHI occurrence depends on the law of closure of two parallel lines. Consequently, the VHI is created when the laws of closure and common fate hold simultaneously.

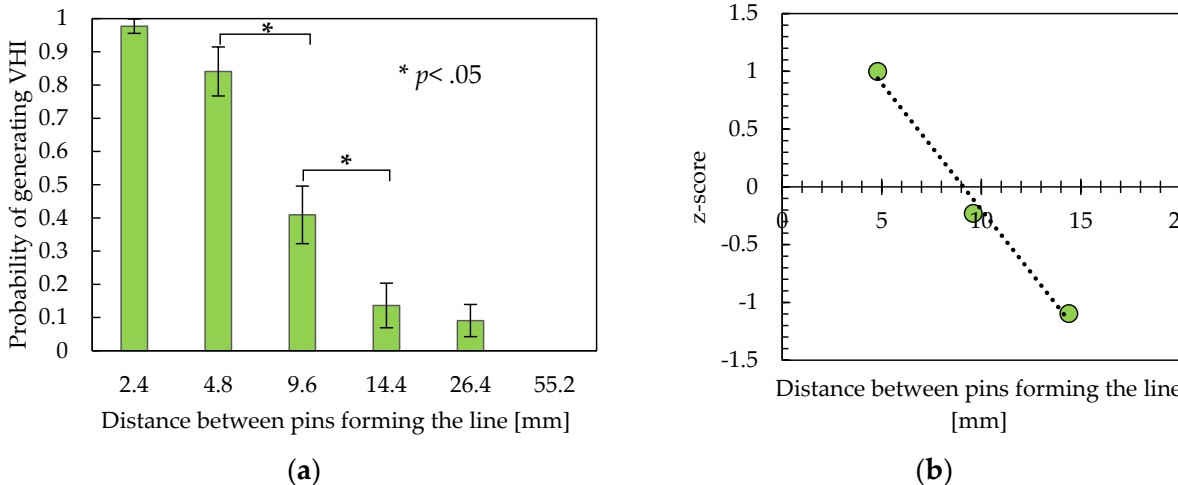

**Figure 7.** Relationship between pin distances when forming line and probability of generating VHI: (**a**) probability of generating VHI and (**b**) z-score.

## 5. Formulation of VHI Using Tactile Gestalt Theory

It was confirmed that the VHI is a tactile Gestalt phenomenon caused by the laws of closure and common fate, which are factors of the principle of Prägnanz. The Gestalt theory has been used in creating designs for the visual sense and melodies for the auditory

sense. Thus, if the tactile Gestalt can be formulated, a texture sensation can be designed, which is the basis for developing the haptic display shown in Figure 8.

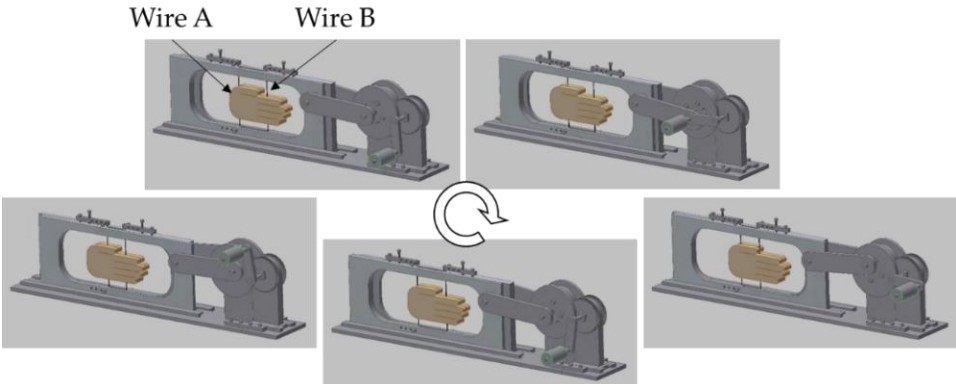

**Figure 8.** Images of movements of two parallel wires with phase difference of 180°.

*5.1. Formulation using Gestalt Theory*

The relationship between the amount of VHI and the physical stimuli caused by two parallel wire movements was utilized to formulate the tactile Gestalt. To realize this formulation, the basic idea is expressed as follows:

$$\psi = f(\phi),\tag{1}$$

where $\psi$ is the psychological quantity, and $\phi$ is the physical stimulus. Therefore, a formulation was developed in which the intensity of the VHI was the psychological quantity and the two parallel wire movements were the physical stimuli.

In the Gestalt theory, the relationship between the parts is important for grouping. Therefore, the phase difference between the two wires with reciprocating movements was selected as the physical stimulus. When two parallel wires reciprocate with a phase difference, the closed area surrounded by the two wires is expanded, contracted or translated. Thus, the law of closure was divided into new sub-factors such as the factors of translation and elasticity. Since two parallel wires oscillate with a simple harmonic oscillation with angular velocity $\omega$ and amplitude $r$, the velocities of wires A and B can be described as follows:

$$v_a = \omega r \sin \omega t,\tag{2}$$

$$v_b = \omega r \sin(\omega t + \phi'),\tag{3}$$

where $\phi'$ is the psychometric phase difference between the motions of A and B.

First, the translation factor is described. It was assumed that the VHI intensity was proportional to the area surrounded by the two lines [9] when they moved in the same direction. Therefore, $G_1$ is designated as the intensity of the VHI generated by the factor of translation:

$$G_1 = a_1 \int_0^{2\pi/\omega} |v_a + v_b| dt + b_1 D,\tag{4}$$

where $D$ is the initial distance between the two parallel wires. The VHI intensity is proportional to wire distance $D$ [9]. $a_1$ and $b_1$ are the coefficient of translation from the distance and the coefficient of $D$ to the VHI intensity, respectively. By substituting Equations (2)–(4), we obtain:

$$G_1 = 4a_1 r (1 + \cos \phi') + b_1 D.\tag{5}$$

Second, the elasticity factor is described. $G_2$ is defined as the intensity of the VHI generated by the elasticity factor. The elasticity factor acts when two parallel wires move in opposite directions and is expressed as follows:

$$G_2 = a_2 \int_0^{2\pi/\omega} |v_a - v_b| dt + b_2 D. \tag{6}$$

By substituting Equations (2) and (3) into Equation (7) we obtain:

$$G_2 = 4a_2 r (1 - \cos \phi') + b_2 D. \tag{7}$$

Since $\phi'$ is the psychometric phase difference, the psychometric quantity can be converted to the physical quantity. Then, the equations of Fechner and Stevens are used [2], which means that the power of the physical quantity is proportional to the psychological quantity as follows:

$$\phi' = c\phi^d. \tag{8}$$

By substituting Equation (8) into Equations (5) and (7), we obtain:

$$G_1 = 4a_1 r \left\{ 1 + \cos\left( c_1 \phi^{d_1} \right) \right\} + b_1 D, \tag{9}$$

$$G_2 = 4a_2 r \left\{ 1 - \cos\left( c_2 \phi^{d_2} \right) \right\} + b_2 D. \tag{10}$$

Equations (9) and (10) show the relationship between the psychological and physical quantities when they are factors of the translation and elasticity, respectively. $(c_1, d_1)$ and $(c_2, d_2)$ are the coefficients of each factor and depend on the individual.

### 5.2. Experimental Setup

To verify the formulation, a psychophysical experiment was conducted, and the experimental value was compared to the value calculated using the formulation. An experimental setup was developed that could provide the participants with two reciprocating parallel wire movements with different phases. This device consisted of an acrylic frame. Each wire moved back and forth because the rotational motion caused by the experimenter turning a lever was converted to translational motion by the crank mechanism. The velocities of the wires are described by Equations (2) and (3). The distance between the wires, $D$, was 70 mm when the phase difference was 0°, and the amplitude, $r$, of each wire was 60 mm. Therefore, the distance between the wires varied from 10 to 130 mm. The rotational speed, $\omega$, was constant at 4.33 rad/s.

### 5.3. Experimental Condition

In this case, 10 types of phase difference conditions were prepared: 0°, 20°, 40°, 60°, 80°, 100°, 120°, 140°, 160° and 180°. The participants evaluated the smoothness of the VHI on a scale from 1 to 7 based on the modulus in which the smoothness of a real velvet fabric corresponded to a score of 7. In one trial, 10 types of phase-difference stimuli were randomly provided to the participants. Each stimulus was provided for 20 s, and a 40 s rest was provided between stimuli. All the trials were performed 10 times by each participant.

### 5.4. Participants

In this case, 10 male Japanese students participated in the psychophysical experiments. They were paid for their participation. Their ages ranged between 22 and 25 yr. All the tests were approved by the ethics committee of the Nagoya University.

### 5.5. Experimental Results and Discussion

Figure 9a shows the relationship between the VHI magnitude and phase difference, obtained from the experimental results. The maximum VHI intensity was found at 0°, with the VHI intensity gradually decreasing when moving toward a phase difference of 100°.

However, after 100°, the VHI intensity gradually increased and a small local maximum was identified at 140°. An ANOVA was conducted to compare the magnitudes using SPSS version 16.0. The phase difference had a significant effect among the 10 conditions ($F$ [2.413, 21.716] = 5.387, $p$ = 0.009). It was assumed that two types of systems were in operation to invoke the VHI, and this result matched our hypothesis. Subsequently, to verify the suggested formulation in which the VHI was invoked under the two factors of elasticity and translation, the experimental results were compared with the predicted values calculated using Equations (9) and (10) with the following parameters. The fitting results are shown in Figure 9b.

$$r = 30 \text{ mm}, \ D = 70 \text{ mm} \tag{11}$$

$$a_1 = 0.0108 \ [1/\text{mm}], \ b_1 = 0.030 \ [1/\text{mm}], \ c_1 = 1.45, \ d_1 = 0.4 \tag{12}$$

$$a_2 = 0.0083 \ [1/\text{mm}], \ b_2 = 0.023 \ [1/\text{mm}], \ c_2 = 1.1, \ d_2 = 1.1 \tag{13}$$

these parameters were the specific values for the participants in this experiment, with different results obtained for different groups.

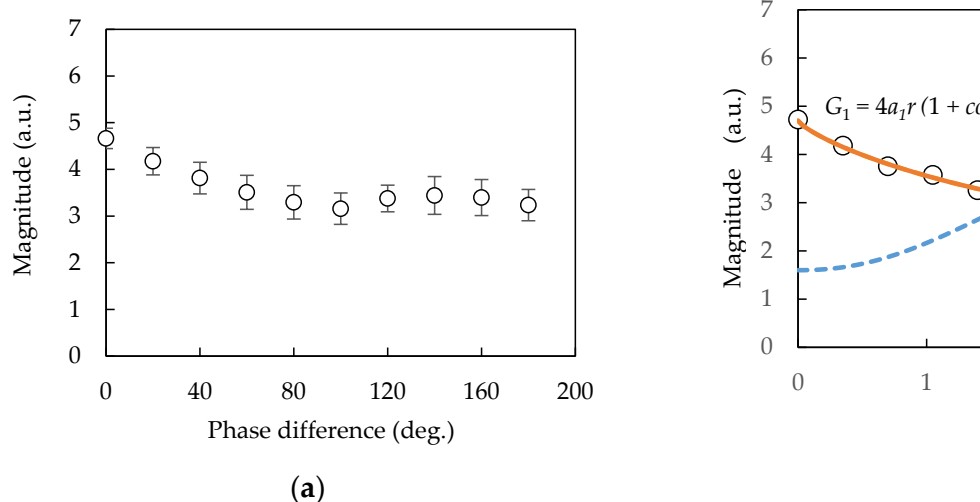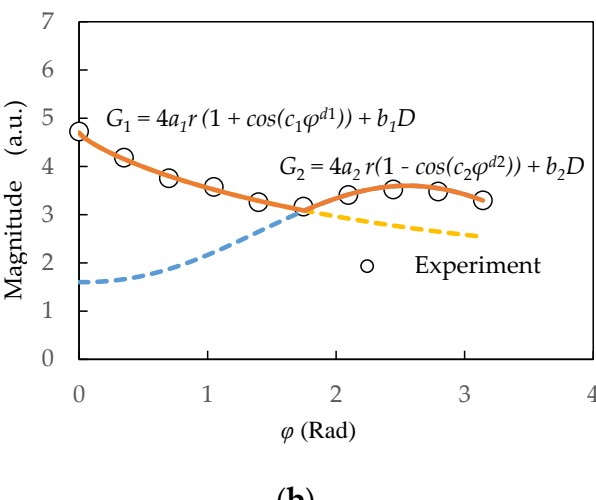

(**a**)          (**b**)

**Figure 9.** VHI intensity for each phase difference between two parallel moving wires and formulation verification. (**a**) VHI intensity and phase difference, (**b**) Estimation of formulated tactile Gestalts factors.

The most important rule of the predicted value calculated using the Gestalt is described in Equation (14). This mathematical formula indicates that a stronger factor is chosen as the Gestalt factor. This feature coincides with the visual and auditory Gestalt rules.

$$VHI = \max(G_1, G_2). \tag{14}$$

Figure 10 shows the competitive relationship between the laws of closure and proximity. When the law of closure is stronger than the law of similarity, the parts that make up the closed shape are recognized as a group (Figure 10a). However, when the law of similarity is stronger than the law of closure, parts with the same color are grouped (Figure 10b). Even though there are individual differences in the specific distances and the positional relationship of the parts related to the victory or defeat of the two factors, this phenomenon is observed in many individuals. Furthermore, the competition between these multiple factors causes visual illusions such as My Wife and My Mother-In-Law [27] or Rubin's goblet-profile [28].

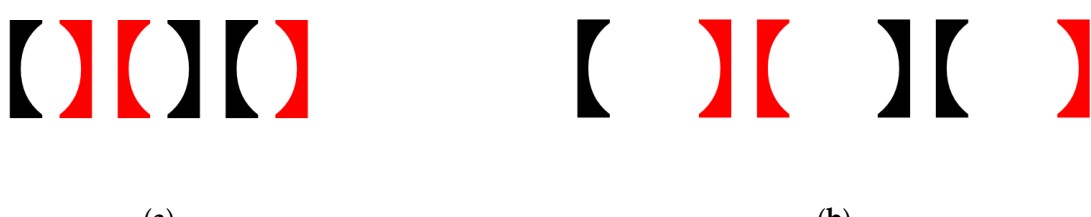

(**a**)  (**b**)

**Figure 10.** Competitive relationship between law of closure and law of proximity. (**a**) Law of closure is stronger than law of similarity, (**b**) Law of similarity is stronger than law of closure.

Therefore, visual studies have clarified that the strong Gestalt factor is prioritized and grouped among the competing Gestalt factors, and this rule might be applied to the sense of touch. To summarize the experimental results, the VHI formulation performed using the laws of elasticity and translation, which were the new Gestalt factors, not only expressed the experimental results highly accurately but also met the rules of the visual Gestalt. Consequently, the validity of the proposed tactile Gestalt concept was confirmed.

## 6. Future Prospects for Tactile Gestalt

This study examined the existence of a tactile Gestalt. Additionally, an attempt was made to formulate the VHI using the principle of Prägnanz. It is noteworthy that the tendency of grouping changes depending on the components that make the Gestalt, and the effect on human sensibilities changes as well. For example, with regard to vision, a web design can be recognized as pleasant or unpleasant with only a slight change in the layout. With regard to hearing, sounds with close frequencies are grouped and recognized as a rhythm, and an individual's mood depends on major or minor keys. It would be interesting if the same effect could be confirmed in the tactile sense. Moreover, the comfortableness produced by a massage is affected by how the therapist moves their fingers and how they apply force against the skin. Thus, the combination of finger movements and forces varying over time affects the comfort. Suzuki et al. attempted to make this series of finger movements and forces into a modified musical score so that the massage could be reproduced at any time [4]. The present study introduced an illusion phenomenon in which a smooth sensation, which was different from the actual contact against the object, was caused when multiple stimuli were grouped using a certain rule. An auditory Gestalt study may be helpful in considering these tactile Gestalt applications. The three features of an auditory Gestalt grouping are the melody, which refers to the connection of frequencies in a time series; tempo, which refers to the strength and speed of the sound; and harmony, which refers to the coordination of chords. Therefore, various musical compositions that are sensed as pleasing contain combinations of these three factors. In addition, it might be interesting to combine the functions of various sensory receptors and present a sense of touch with a Gestalt grouping, such as a Gestalt of a temperature sensation, pain sensation, vibration sensation and shearing force. Thus, work will be carried out to develop a haptic display that provides people with the above stimuli simultaneously and investigate the effect of these combinations. It is expected that the feeling of touch is induced with respect to space and emotion is induced with respect to time in the tactile Gestalt. In the future, studies will be conducted on the tactile sense with goal of promoting a new haptic theory. This research finding will advance the field of tactileology.

## 7. Conclusions

To develop an advanced haptic display that manipulates people's tactile sensation as well as sensibilities, a time-varying tactile sensation in the form of the tactile illusion phenomenon was considered, and some tactile illusion mechanisms were described using the rules of visual and auditory illusion mechanisms. To formulate a theory on the tactile illusion, Gestalt grouping was considered. Gestalt grouping is a mechanism in which multiple stimuli are integrated into the central nervous system to invoke sensations. The

research on visual Gestalts has been particularly advanced, and the tendency of grouping is summarized by the principle of Prägnanz. Based on this principle, some studies have also investigated the auditory Gestalt, which is grouped on the time axis. Since some visual or auditory illusions are caused by the Gestalt Grouping mechanism, it was assumed that tactile illusion phenomena are also caused by Gestalt grouping. This study attempted to investigate the operation of a tactile Gestalt, focusing on the induction mechanism of the VHI. Two types of psychophysical experiments were conducted. First, an investigation was conducted on the relationship between the laws of closure and common fate, which are the principles of Prägnanz and the occurrence of the VHI. Second, the VHI intensity was formulated using the stimulus and principle of Prägnanz. Consequently, it was confirmed that the VHI is induced in the tactile Gestalt when the laws of closure and common fate are satisfied. Furthermore, it was clarified that the VHI could be formulated with the tactile principle of Prägnanz. This finding suggested that a tactile Gestalt causes a tactile sensation in space, which changes with respect to time. Currently, Gestalt research is limited to using the two-line stimulation. However, in the future, various tactile receptor characteristics will be incorporated, such as cold, warm and pain sensations, and the characteristics of the tactile Gestalt will be clarified.

**Author Contributions:** Conceptualization, M.O.; methodology and data curation, T.N.; investigation, validation, writing—original draft preparation, writing—review and editing, H.K. All authors have read and agreed to the published version of the manuscript.

**Funding:** This research has been supported by the Kayamori Foundation of Informational Science Advancement, and the JSPS Kakenhi Grants Numbers JP20K23346 and 19K22869.

**Institutional Review Board Statement:** The study was conducted according to the guidelines of the Declaration of Helsinki, and approved by the Ethics Committee of Nagoya University.

**Informed Consent Statement:** Informed consent was obtained from all subjects involved in the study.

**Data Availability Statement:** The data presented in this study are available on request from the corresponding author.

**Conflicts of Interest:** The authors declare no conflict of interest.

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
