# Peer review of "Investigation of Tactile Illusion Based on Gestalt Theory"

_philosophies, doi:10.3390/philosophies6030060_

Round 1

Reviewer 1 Report

To the authors.

There is a general problem with the paper related to communication: who is the reader that the authors are addressing?

The paper starts with referring to very specific problems related to the technical possibility of making tactile displays. Since the authors do not explain at all what these problems (relating to technical limitations of microactuators and cutaneous transduction mechanisms) exactly amount to, the authors lose all non-expert readers from the start. But why then submit this paper to a journal that is not addressing such an expert audience ? Moreover, later on in the text, the authors give a very elementary introduction to Gestalt principles in vision and hearing. So there is a discrepancy between the technicality that is simply assumed in the first sections and the very non-technicality of the later section on Gestalt. Most readers of this journal probably already know everything that is said about Gestalt, but have no idea what piezoelectric elements or electrostatic actuators are. So why do we get an extremely basic explanation of the former (including illustrations of very common examples) and no explanation at all about the latter? So it is not clear why the authors have chosen this journal nor who will benefit from reading this paper.

The authors refer 3 times to tactileology (including in the very first sentence), without in any way explaining what this is.

And there is brief reference to massage techniques; how does this relate to tactile displays?

There is a lot of repetition. What VHI and Gestalt is, is explained several times anew.

The authors first state that previous research has shown that the optimal condition for the illusion is two parallel wires. Then, the authors want to argue that VHI involves Gestalt grouping by pointing out that the effect does not occur when there is only a single wire. However, in that very sentence they say it requires “more” than 2 wires, while they had just said that the optimal condition requires 2 wires.

The author(s’) writing is at times imprecise:

-They say that illusions are caused by Gestalt Theory. A theory does not cause illusions.

-The definitions of Gestalt are imprecise. Maybe it is better to quote a standard definition from a historical authority on the matter.

-(there are typos in Müller-Lyer illusion and the name of Ehrenfels)

The main problem with the paper is the lack of systematic presentation of the context in which this research is situated. In my opinion the authors should find a way of ordering the material in more systematic way: for example, first say what the framework is (which discipline, which field of applied science, etc), what the broader goal is, then a (well-written) explanation of Gestalt principles (only the relevant ones), then why tactile Gestalt is a still in need of exploration, why exactly this matters (what is the purpose and so on). As it stands, all these things are mixed up and usually stated in unclear writing.

The material is interesting, no doubt. But the presentation is detracting from the content.

Author Response

For Reviewer #1

Thank you for your comments. We fixed the problems that you pointed out and had a native person check the manuscript. Regarding the revisions, the main revision is written in red letters and the minor revision is written blue letters.

Q1: There is a general problem with the paper related to communication: who is the reader that the authors are addressing?

A1: As you pointed out, I did not consider the expertise of the readers of this treatise. We address the tactile researchers including a philosopher and an engineer as a reader. However, as you pointed out, we only described the actuators problems as the tactile research issue in introduction section. However, this will not attract our addressing readers. Thus, we improved a problem presentation from the specific engineering issue to the general issue as follows.

Before: To progress the tactileology, it is necessary to investigate not only the tactile sense but also the effects of tactile sensation on human psychology. To reveal the former, studies of ideal haptic display have been conducted and studies on the role of tactile sensation have also been conducted. However, an ideal tactile display has not been realized because of the difficulty in fabricating microactuators which are suitable for mechanoreceptors of the skin. Moreover, it is difficult to perceive texture sensation with a haptic display owing to the complexity of human tactile sensing systems. To address these problems, many actuators have been developed using piezoelectric elements [1], shape memory alloys [2], polymer actuators [3], pneumatic actuators [4], and electrostatic actuators [5]. However, it is difficult to realize a texture sensation to an operator using a haptic display since the actuator size has not reached the idea size.

After: The human tactile sense plays an important role not only in sensing the external environment but also in communication. Tactile researchers have focused on the former, including efforts to understand how humans detect an object in their environment and perceive it as a subjective tactile sensation. In this research, various types of actuators have been developed for tactile displays [1]. In addition, different psychophysical experimental methods for evaluating human subjective sensations have been developed [2], along with some mathematical formulas to describe the relationship between psychological quantities and physical quantities, such as Stevens' power law and the Weber–Fechner law. On the other hand, there has been little systematic research on communication via the tactile sense. When considering communication via the sense of touch, it seems necessary to consider the time evolution of the tactile sensation. Thus, the term “tactileology” was suggested by Suzuki to define the new research field of time-evolutional tactile sensation [3]. To advance this research field, this study attempted to clarify the effects of a time-varying input on a human.

Q2: The paper starts with referring to very specific problems related to the technical possibility of making tactile displays. Since the authors do not explain at all what these problems (relating to technical limitations of microactuators and cutaneous transduction mechanisms) exactly amount to, the authors lose all non-expert readers from the start. But why then submit this paper to a journal that is not addressing such an expert audience? Moreover, later on in the text, the authors give a very elementary introduction to Gestalt principles in vision and hearing. So there is a discrepancy between the technicality that is simply assumed in the first sections and the very non-technicality of the later section on Gestalt. Most readers of this journal probably already know everything that is said about Gestalt, but have no idea what piezoelectric elements or electrostatic actuators are. So why do we get an extremely basic explanation of the former (including illustrations of very common examples) and no explanation at all about the latter? So it is not clear why the authors have chosen this journal nor who will benefit from reading this paper.

A2: As you pointed out, there is a discrepancy between the technicality of the actuators and the non-technicality on Gestalt in introduction. Since we were supposed to provide this paper to all tactile researchers including an engineer and a philosopher, we changed the description of actuator issue to the basic conventional tactile research issue. The time-evolving tactile sensation was not investigated even though we usually have a communication with animals as well as people via time-evolving tactile sense. Thus, we revised the introduction as follows.

Before: However, an ideal tactile display has not been realized because of the difficulty in fabricating microactuators which are suitable for mechanoreceptors of the skin. Moreover, it is difficult to perceive texture sensation with a haptic display owing to the complexity of human tactile sensing systems. To address these problems, many actuators have been developed using piezoelectric elements [1], shape memory alloys [2], polymer actuators [3], pneumatic actuators [4], and electrostatic actuators [5]. However, it is difficult to realize a texture sensation to an operator using a haptic display since the actuator size has not reached the idea size.

After: On the other hand, there has been little systematic research on communication via the tactile sense. When considering communication via the sense of touch, it seems necessary to consider the time evolution of the tactile sensation. Thus, the term “tactileology” was suggested by Suzuki to define the new research field of time-evolutional tactile sensation [3]. To advance this research field, this study attempted to clarify the effects of a time-varying input on a human.

Q3: The authors refer 3 times to tactileology (including in the very first sentence), without in any way explaining what this is.

A3: As you pointed out, we did not describe the definition of tactileology even though this word is an important word of the special issue of this journal. This new word is defined as a new cross-disciplinary research field that considers time-evolving tactile sensations. The conventional tactile research only dealt with the human’s subjective sensation, such as “hard" and "soft” without changing with time. However, the time-evolving touch is also an important factor for communication between people or animals. Therefore, we added the description of the tactileology to the abstract and introduction as follows.

[Abstract]

Time-evolving tactile sensations are important in communication between animals as well as humans. In recent years, this research area has been defined as “tactileology,” and various studies have been conducted.

[Introduction]

Thus, the term “tactileology” was suggested by Suzuki to define the new research field of time-evolutional tactile sensation [3]. To advance this research field, this study attempted to clarify the effects of a time-varying input on a human.

Q4: There is brief reference to massage techniques; how does this relate to tactile displays?

A4: We are sorry for not made it clear enough in the previous manuscript. The tactile display is a device that provides someone with mechanical vibrations, electrical stimulation and so on for any purposes such as VR or massage. Thus, there are various type displays depending on the application. I believe that if massage techniques are translated to the vibration data with the tactile score, the massage can be reproduced with the tactile display with vibrational stimulus.

Q5: There is a lot of repetition. What VHI and Gestalt is, is explained several times anew.

A5: As you pointed out, I removed the repeated description about what is VHI and Gestalt.

A5-1: We removed the following VHI description in section 3.

Velvet hand illusion (VHI) is a tactile illusion phenomenon in which a smooth sensation is induced when two parallel wires (lines) are moved back and forth (reciprocated) across the palms of a human’s hands. This illusion is known to occur in most people and is exhibited at the Exploratorium in San Francisco, California, USA.

A5-2: We removed the following sentence at the beginning of the section 4.5.

We assumed that the VHI is a tactile Gestalt phenomenon.

Q6: The authors first state that previous research has shown that the optimal condition for the illusion is two parallel wires. Then, the authors want to argue that VHI involves Gestalt grouping by pointing out that the effect does not occur when there is only a single wire. However, in that very sentence they say it requires “more” than 2 wires, while they had just said that the optimal condition requires 2 wires.

A6: As you pointed out, our explanation was not correct due to my misunderstanding about the English expression. we assumed that “more than 2” includes “2”. That is why, in this sentence, what we were trying to say was that VHI occurs in more than one wire, and the condition of two parallel wires is optimal. We revised the manuscript as follows.

Before: The VHI is not generated under a one-wire condition but is generated under the condition of more than two wires [9][10]

After: The VHI is not generated under a one-wire condition but is generated under the condition of more than one wire [9][10]

Q7: They say that illusions are caused by Gestalt Theory. A theory does not cause illusions.

A7: As you pointed out, there was a contradiction in the sentence. We corrected the expression from “They say that illusions are caused by Gestalt Theory.” to “They say that illusions are caused by Gestalt Grouping mechanism.”

Q8: The definitions of Gestalt are imprecise. Maybe it is better to quote a standard definition from a historical authority on the matter.

A8: As you pointed out, we improved the definitions of Gestalt based on the standard definition from a historical authority [Willis, 1999].

Reference

  1. D. Willis, A Source Book of Gestalt Psychology, Taylor & Francis, 1999.

Before: Here, the German word “Gestalt” translates to shape. Gestalt is not a single part but refers to multiple parts of a shape and is different from simple combinations of parts.

After: According to “A source book of gestalt psychology” [7], the fundamental formula of the Gestalt theory might be expressed as follows: there are wholes, the behavior of which is not determined by that of their individual elements, but where the part-processes are themselves determined by the intrinsic nature of the whole. In other words, the wholes cannot be expressed by a simple sum of elements, and elucidating the mechanism by which the elements are integrated is the research objective.

Q9: (there are typos in Müller-Lyer illusion and the name of Ehrenfels)

A9: Thank you very much for letting me know. we corrected the misspelling as follows: Müller-Lyre illusion to Müller-Lyer illusion and Ehrenfel to Ehrenfels.

Q10: The main problem with the paper is the lack of systematic presentation of the context in which this research is situated. In my opinion the authors should find a way of ordering the material in more systematic way: for example, first say what the framework is (which discipline, which field of applied science, etc), what the broader goal is, then a (well-written) explanation of Gestalt principles (only the relevant ones), then why tactile Gestalt is a still in need of exploration, why exactly this matters (what is the purpose and so on). As it stands, all these things are mixed up and usually stated in unclear writing.

A10: Based on the pointed out, I revised the introduction and abstract to make it more systematic.

Q10-1: What the framework is (which discipline, which field of applied science, etc):

A10-1: We aimed to construct a basic theory for an engineering application of tactile sense. For that purpose, it is necessary to elucidate the process of integrating multiple tactile stimuli. To elucidate the integration process, it is necessary to observe and formulate the phenomena experienced by humans. Therefore, the framework used in this study refers to human perceptual law of organization, such as Gestalt theory. To achieve this goal, we tried to formulate the relationship between the time-evolving tactile stimuli (physical quantities) and subjective tactile sensations (psychological quantities). Since a word of framework is very ambiguous, I changed this word to Gestalt theory.

Q10-2: What the broader goal is:

A10-2: The border goal is to elucidate the effect of the time-evolving tactile sensation (tactileology) on human’s body and mind such as massage and humanitude. Furthermore, in the future, we develop a tactile display that have a positive impact on the human mind and body.

[Introduction: 42-44]

It is very interesting to note that various stimuli that change over time, especially tactile stimuli, relieve stress or change a person’s mood. We believe that elucidating this mechanism will be important for humans living in a stressful society.

[Introduction: 67-68]

A philosophical goal of this study was to clarify tactileology using the tactile Gestalt theory.

Q10-3: a (well-written) explanation of Gestalt principles (only the relevant ones)

A10-3: Since we did not use the law of proximity in experiment and discussion, I removed the description of law of proximity in section 2.2.

Q10-4: Why tactile Gestalt is a still in need of exploration:

A10-4: The science and technology are good at investigating the component technology. However, this is not good at investigating the integration of the component technology, since the sum of them do not express the characteristics of the whole. Therefore, we need to know the process how to integrate them by observing the phenomena. Regarding the human, Gestalt theory is the integration formula of some sensory inputs. That is why, we tried to formulate the tactile sensation based on the tactile Gestalt theory.

[Introduction: 55-58]

In this way, the Gestalt theory can deal with the mechanism for integrating many elements, which current science and technology are not good at handling. For this reason, it was assumed that this theory could also handle the integration of tactile sensations that change over time.

Reviewer 2 Report

The paper presents two studies of tactile experience that aim to assess the use of Gestalt Theory for tactile sensations, extending this Gestalt framework beyond vision and sound.

I read the paper in full, but already reading the abstract it was clear that the paper needs to be rewritten as the English was not sufficient to make a really clear paper. It is essential that someone with a good grip on English goes through the manuscript and improves phrasing and clarity in general. Also introducing terms and technical details could be much improved.

In addition to this problem, I find this paper not (yet) very strong as a philosophy paper. The experimental setup looks good and the topic is very interesting (being a philosopher I can only judge the soundness of these experiments at a superficial level). However, the conceptual and philosophical implications are less developed, also due to the language issue, I suppose.

I would suggest (in addition to a language revision) to provide a more extensive introduction of the aim of this paper, an introduction of the Gestalt framework (to enable a wider audience to read the paper), and a clearer philosophical goal for this paper. Now it remains mostly a technical story, which has interesting implications for perceptual research, but the broader story is not so well developed.

Just to give an idea about the reasons for the judgment above, here are some examples drawn from the text.

“actuator size has not reached the idea size.” What does this mean, “idea size”?

“Considering the role of communication tools, scholars” What does this mean and how is this relevant?

“We hypothesize the effects of time-varying tactile inputs on human psychology.” Clarify. ‘Psychology’ is too general a term as the target of specific inputs, so perception or even mind makes more sense here.

“pseudo-haptics” In philosophy it is not a common practice to leave terms unexplained and only use a reference.

“two parallel wires (lines) move reciprocated on palm of hand.” This usage of ‘reciprocation’ is in the dictionary but it is not standard usage. It needs to be introduced to help the reader understand. Maybe: “two parallel wires (lines) are moved back and forth (reciprocated) across the palms of a human’s hands” Or alternatively: “two parallel wires (lines) are reciprocated (moved back and forth) across the palms of a human’s hands”

Author Response

For Reviewer #2

Thank you for your comments. We fixed the problems that you pointed out and had a native person check the manuscript. Regarding the revisions, the main revision is written in red letters and the minor revision is written blue letters.

Q1: The paper presents two studies of tactile experience that aim to assess the use of Gestalt Theory for tactile sensations, extending this Gestalt framework beyond vision and sound. I read the paper in full, but already reading the abstract it was clear that the paper needs to be rewritten as the English was not sufficient to make a really clear paper. It is essential that someone with a good grip on English goes through the manuscript and improves phrasing and clarity in general. Also introducing terms and technical details could be much improved.

Q2: In addition to this problem, I find this paper not (yet) very strong as a philosophy paper. The experimental setup looks good and the topic is very interesting (being a philosopher I can only judge the soundness of these experiments at a superficial level). However, the conceptual and philosophical implications are less developed, also due to the language issue, I suppose.

A1 & A2: At first, we revised abstract and introduction to make it more systematic. Second, we requested native speakers of English to proofread our English writing. Third, as you pointed out, since the conceptual and philosophical implications were less developed, we removed the description of an engineering issue from the abstract and introduction and focused on the description of tactile perception processing and tactile Gestalt theory. We will explain the details by answering the following questions.

Q3: I would suggest (in addition to a language revision) to provide a more extensive introduction of the aim of this paper, an introduction of the Gestalt framework (to enable a wider audience to read the paper), and a clearer philosophical goal for this paper. Now it remains mostly a technical story, which has interesting implications for perceptual research, but the broader story is not so well developed.

Q3-1: language revision

A3:1: We requested native speakers of English to proofread our English writing.

Q3-2: To provide a more extensive introduction of the aim of this paper:

A3-2: Based on your pointed out, we added the general problem of the current tactile research and we removed the content unrelated to this paper, such as the actuator issue.

Q3-3: An introduction of the Gestalt framework (to enable a wider audience to read the paper):

A3-3: The word of framework is very ambiguous for readers. We assumed that the word of Gestalt framework is equivalent to the principle of Prägnanz. That is why, we removed the Gestalt framework from this paper and we explained the principle of Prägnanz and Gestalt theory in detail in an introduction section.

Q3-4: A clearer philosophical goal for this paper:

A3-4: A clearer philosophical goal is to develop a theory that can explain a new discipline called Tactileology, which is the theme of the special issue of this Journal. As a method, we tried to formulate the effect of time-varying tactile sensation on human subjective evaluation. We added the following sentence in an introduction section.

A philosophical goal of this study was to clarify tactileology using the tactile Gestalt theory.

Q4: Just to give an idea about the reasons for the judgment above, here are some examples drawn from the text. “actuator size has not reached the idea size.” What does this mean, “idea size”?

A4: We are very sorry for typo and thanks for pointing it out. Here, We were supposed to write “ideal” tactile display. “Ideal” means that the tactile display ability matches human tactile sensory properties. Taking into consideration of the reviewer #1’s comment, the description of actuators was not suitable for the aim of this paper since our experiment was not related to the actuators. That is why, we removed this description from this paper.

Q5: “Considering the role of communication tools, scholars” What does this mean and how is this relevant?

A5: As you pointed out, this sentence is too ambiguous for the readers to understand. In this sentence, we were supposed to suggest that not only between humans but also between animals have a communication via tactile sense. Furthermore, in this tactile communication, the time-evolving tactile sensation plays an important role. The text was revised to make it easier for readers to understand.

Before: Considering the role of communication tools, scholars have focused on studying the effects of time-varying input on the tactile sense. To this end, various technologies have been developed, such as the tactile score [6] and humanitude [7].

After: On the other hand, there has been little systematic research on communication via the tactile sense. When considering communication via the sense of touch, it seems necessary to consider the time evolution of the tactile sensation. Thus, the term “tactileology” was suggested by Suzuki to define the new research field of time-evolutional tactile sensation [3]. To advance this research field, this study attempted to clarify the effects of a time-varying input on a human.

A tactile score [4] and humanitude [5] are some of the techniques currently in use that utilize a time-varying tactile sensation.

Q6: “We hypothesize the effects of time-varying tactile inputs on human psychology.” Clarify. ‘Psychology’ is too general a term as the target of specific inputs, so perception or even mind makes more sense here.

A6: As you pointed out, ‘Psychology’ is too general a term and we had to use the other concrete term. In this sentence, what we were trying to say is that time-varying tactile inputs affects the human’s body health (massage) as well as the mental health (humanitude). That is why, we corrected this expression of human psychology to human’s body and mental health in section 1.

Q7: “pseudo-haptics” In philosophy it is not a common practice to leave terms unexplained and only use a reference.

A7: As you pointed out, pseudo-haptics is not common practice. Not only pseudo-haptics but also the other illusion which were described in an introduction section made our paper understand difficult. That is why, we removed the description of various tactile illusion phenomena from this paper.

Q8: “two parallel wires (lines) move reciprocated on palm of hand.” This usage of ‘reciprocation’ is in the dictionary but it is not standard usage. It needs to be introduced to help the reader understand. Maybe: “two parallel wires (lines) are moved back and forth (reciprocated) across the palms of a human’s hands” Or alternatively: “two parallel wires (lines) are reciprocated (moved back and forth) across the palms of a human’s hands”

A8: As you pointed out, the VHI explanation is very wrong. we revised this expression as follows based on your advice.

Before: two parallel wires move reciprocated on palm of hand

After: two parallel wires are moved back and forth across the palms of a human’s hands

Round 2

Reviewer 1 Report

In my opinion, the presentation of the framework at the beginning of the paper has much improved. 

It is still hard to see why the authors write that the clarification of tactileology is "a philosophical goal".

It might also be usueful to the authors to look at philosophy. In their response to the referee's questions, the authors write:

"This new word is defined as a new cross-disciplinary research field that considers time-evolving tactile sensations. The conventional tactile research only dealt with the human’s subjective sensation, such as “hard" and "soft”  without changing with time."

The suggestion that other research has only dealt with non-temporal sensations (like hard and soft) is not really true, at least not for philosophy. In a recent article (that was not specifically about touch), a couple of sources are listed that focus on the tactile perception of "events". They refer to Richardson 2011 "Bodily Sensation and Tactile Perception" in PPR; Brogaard 2012; "What do we say when we say how we feel"; and Mattens 2017 "The Sense of Touch: From Tactility to Tactual Probing" in Australasian Journal of Philosophy. In particular the last one entirely focuses on sensing time-evolving events as necessary for sensing properties like "hard" and "soft".

Author Response

For reviewer #1

Thank you very much for pointing out the part of our insufficient explanation and providing us the several articles relating to the tactile perception. Based on your comments, we improved the descriptions of philosophical goal and tactileology as below.

Q1: It is still hard to see why the authors write that the clarification of tactileology is "a philosophical goal".

A1: As you pointed out, there was a lack of description why the clarification of tactileology is "a philosophical goal". Tactileology is an interdisciplinary research field and it focuses on the relationship between the time-evolving skin deformation and not only the perception of object property such as “soft” and “hard” but also the sensibility such as “comfortableness”. Thus, when we clarify the tactileology, we think that the understanding of fundamental nature of human’s tactile sense will be improved. Furthermore, since philosophy is the study of the fundamental nature including human, we consider that the clarification of tactileology is "a philosophical goal".

We improved the tactileology’s description in the introduction section as below.

(before)

In this research, a philosophical goal of this study was to clarify tactileology using the tactile Gestalt theory.

(after)

In this research, a philosophical goal of this study was to clarify tactileology using the tactile Gestalt theory. This is because the clarification of tactileology was considered to contribute towards the understanding of fundamental nature of human’s tactile sense.

Q2: It might also be useful to the authors to look at philosophy. In their response to the referee's questions, the authors write:"This new word is defined as a new cross-disciplinary research field that considers time-evolving tactile sensations. The conventional tactile research only dealt with the human’s subjective sensation, such as “hard" and "soft” without changing with time." The suggestion that other research has only dealt with non-temporal sensations (like hard and soft) is not really true, at least not for philosophy.

In a recent article (that was not specifically about touch), a couple of sources are listed that focus on the tactile perception of "events". They refer to Richardson 2011 "Bodily Sensation and Tactile Perception" in PPR; Brogaard 2012; "What do we say when we say how we feel"; and Mattens 2017 "The Sense of Touch: From Tactility to Tactual Probing" in Australasian Journal of Philosophy.

In particular, the last one entirely focuses on sensing time-evolving events as necessary for sensing properties like "hard" and "soft".

A2: We confirmed the recent articles which you provided us. As you pointed out, when human perceives a characteristic of object such as “hard" and "soft”, the sensing time-evolving events is essential. That is why, when we describe the tactileology, we have to explain how tactileology differs from the above-mentioned conventional research. Tactileology focuses on the relationship between the time-evolving skin deformation and not only the perception of object property but also the sensibility such as “comfortableness”. Thus, we modified the description about tactileology as below.

(before)

Thus, the term “tactileology” was suggested by Suzuki to define the new research field of time-evolutional tactile sensation [10]. To advance this research field, this study attempted to clarify the effects of a time-varying input on a human.

(after)

Thus, the term “tactileology” was suggested by Suzuki to define the new interdisciplinary research field of time-evolutional tactile sensation [10]. Tactileology focuses on the relationship between the time-evolving skin deformation and not only the perception of object property but also the sensibility such as “comfortableness”. To advance this research field, this study attempted to clarify the effects of a time-varying input on a human.